# Does RoBERTa Perform Better than BERT in Continual Learning: An Attention Sink Perspective

**Xueying Bai, Yifan Sun, Niranjan Balasubramanian**
Department of Computer Science
Stony Brook University
{xubai,ysun,niranjan}@cs.stonybrook.edu

## Abstract

Continual learning (CL) aims to train models that can sequentially learn new tasks without forgetting previous tasks' knowledge. Although previous works observed that pre-training can benefit CL, it remains unclear whether a pre-trained model with higher downstream capacity also performs better in CL. In this paper, we observe that pre-trained models may allocate high attention scores to some 'sink' tokens, such as [SEP] tokens, which are ubiquitous across various tasks. Such attention sinks may lead to models' over-smoothing in single-task learning and interference in sequential tasks' learning, which may compromise the models' CL performance despite their high pre-trained capabilities. To reduce these effects, we propose a *pre-scaling mechanism* that encourages attention diversity across all tokens. Specifically, it first scales the task's attention to the non-sink tokens in a probing stage, and then fine-tunes the model with scaling. Experiments show that pre-scaling yields substantial improvements in CL without experience replay, or progressively storing parameters from previous tasks.

## 1 Introduction

Machine learning applications in the real world often need to face continuous streams of data from different tasks or distributions (Lopez-Paz & Ranzato, 2017; Hou et al., 2019). For such cases, it is important to develop continual learning (CL) models that can progressively learn new tasks without performance degradation in previous tasks (i.e., catastrophic forgetting).

The pre-training and fine-tuning paradigm (Devlin et al., 2018), which effectively learns downstream tasks by fine-tuning a pre-trained language model (LM), is widely used for general NLP tasks. Previous works (Wu et al., 2022; Mehta et al., 2023) observed that pre-training can also benefit CL. However, it remains unclear whether a pre-trained model that has better single-task performance also performs better in CL settings. For example, BERT (Devlin et al., 2018) and RoBERTa (Liu et al., 2019b) are two pre-trained LMs with the same model structure. RoBERTa achieves generally better downstream performance than BERT, in part because it is pre-trained on more diverse data. In CL, however, RoBERTa does not always outperform BERT (Wu et al., 2022). This motivates our study on the factors that may cause models' inferior performance in CL, besides their pre-trained capacity.

In this paper, we show that the *attention sink* phenomenon can influence models' CL capacity. Attention sinks have been observed on autoregressive LLMs, where models tend to allocate high attention scores to specific tokens in the input ('*sink tokens*') regardless of their semantic significance (Xiao et al., 2024). In Fig. 1, we show that attention sinks appear after the first layers in both BERT and RoBERTa models. And the sink tokens are usually common tokens (e.g., special tokens like [SEP]), which are not semantically significant but are present in most NLP tasks (Fig. 3 right). However, unlike the previous work by Xiao et al. (2024) which focuses on the magnitude of attention scores on sink tokens, our work focuses on the attention deviations on sink tokens ('*sink attention deviations*') and their influence on CL.

---

The code is available at: https://github.com/StonyBrookNLP/attention-sink-cl

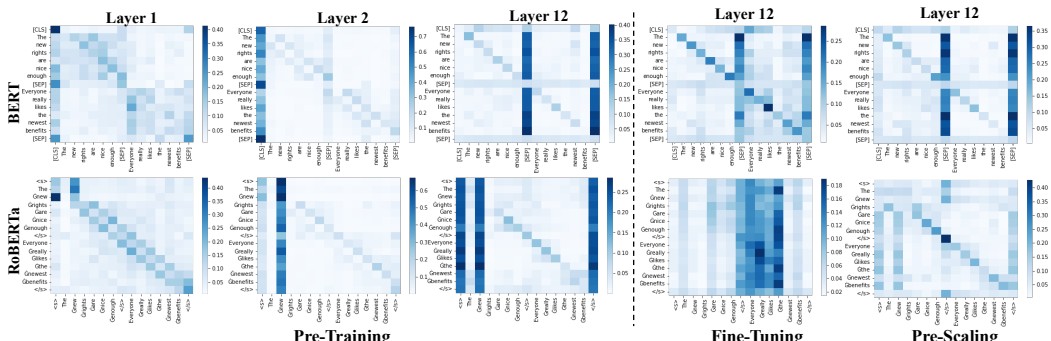

Figure 1: Attention maps averaged from all attention heads after pre-training, fine-tuning, and our pre-scaling mechanism. Sink tokens (i.e., with dark blue columns) in pre-trained models obtain similar high attention scores. After fine-tuning, models (especially RoBERTa) have drastic attention changes, which may indicate feature distortion. After pre-scaling, models have diverse attention on sink tokens and preserve the pre-trained attention patterns.

In particular, we connect the attention deviations on sink tokens to the *over-smoothing* phenomenon. Over-smoothing has been observed in pre-trained LMs, where models output nearly identical representations for all input tokens (Dong et al., 2021; Shi et al., 2022). We show that over-smoothing is related to small sink attention deviations in pre-trained models. It can cause distortions of pre-trained features (Kumar et al., 2022), which may make models less generalizable to out-of-distribution data (e.g., data from other tasks) and affect CL.

Models' small attention deviations on *common* sink tokens can also cause unnecessary interference across tasks in CL. Specifically, representations of common sink tokens may carry the information of one task, which then influences another task's learning. This can be harmful if the new task is irrelevant to the previous one. We conduct a case study to show that such interference is hard to avoid when models have small sink attention deviations.

To address the above problems, we propose a pre-scaling mechanism that encourages diverse attention scores on sink tokens. It introduces a scaling layer that is first learned to allocate diverse attention to tokens in a probing stage, and then tuned in a fine-tuning stage together with the pre-trained encoder. Experiments show that pre-scaling improves models' CL performance with reduced over-smoothing. Moreover, RoBERTa models consistently outperform BERT models after pre-scaling, which suggests that pre-scaling helps to better utilize models' pre-trained capacity in CL.

In conclusion, we make the following contributions: (1) We characterize the attention sink phenomenon in pre-trained LMs and build a connection to over-smoothing. (2) We conduct a case study to show that the above attention sinks may propagate unexpected interference in CL. (3) We propose a pre-scaling mechanism that can significantly improve pre-trained LMs capacity in CL without experience replay or progressively storing parameters.

## 2 Related Work

**Continual Learning.** Models for CL can be divided into three main categories: (1) regularization-based models which constrain the deviation of new parameters from the older ones (Kirkpatrick et al., 2017; Zenke et al., 2017; Aljundi et al., 2018; Lee et al., 2017); (2) replay-based models which reduce catastrophic forgetting by rehearsing on real or pseudo samples from previous tasks (Lopez-Paz & Ranzato, 2017; Chaudhry et al., 2019a) or generative models (Shin et al., 2017; Kemker & Kanan, 2017); (3) architecture-based models which learn evolving architectures for sequential tasks, with their capacities for each task carefully assigned (Rusu et al., 2016; Yoon et al., 2017).

CL in NLP is an emerging area (Liu et al., 2019a; Biesialska et al., 2020). MBPA++ (d'Autume et al., 2019) uses experience replay and local adaptation to mitigate forgetting; LAMOL (Sun et al., 2019) generates pseudo samples for replay; IDBR (Huang et al., 2021a) disentangles

task-agnostic and task-specific information; CTR (Ke et al., 2021) uses a capsule network for knowledge transfer. All the above models are based on pre-trained LM (Devlin et al., 2018; Brown et al., 2020; Raffel et al., 2019). Recent works show that pre-training can alleviate catastrophic forgetting (Wu et al., 2022; Mehta et al., 2023; Lee et al., 2023). Mehta et al. (2023) claims that the benefit may come from having less sharp minima. In this paper, we tackle the CL problem from an attention sink perspective, which provides an explanation of why sometimes RoBERTa underperform BERT in CL tasks (Wu et al., 2022). To the best of our knowledge, we are the first to tackle CL problems from this angle.

**Over-Smoothing.** In this paper, we connect attention sinks to an over-smoothing phenomenon, which is first proposed in graph neural networks (Oono & Suzuki, 2020; Huang et al., 2020; Cai & Wang, 2020; Rusch et al., 2023; Yang et al., 2020). Over-smoothing refers to the problem that the models' performance deteriorates as representations of all the nodes in the graph become similar (Li et al., 2018; Xu et al., 2018). For transformer-based models, Dong et al. (2021) claims that pure attention loses rank doubly exponentially with model depth. And Shi et al. (2022) characterize the oversmoothing problem in transformers by viewing the attention matrix as a form of adjacency matrix in the graph. In this paper, we connect the over-smoothing problem to attention sinks, to show that in some cases attention sinks will influence model's task learning and cause inferior performance in CL.

## 3 Attention Sinks in Language Models

We first empirically show that certain attention sinks exist in pre-trained LM layers. Then we study their impact by connecting them to an over-smoothing phenomenon, which may influence models' single-task overfitting and in turn can influence their CL abilities. The attention sinks can also cause cross-task interference in CL, which we discuss in section 4.

### 3.1 Empirical Analysis of Attention Sinks

We characterize the presence of attention sinks (Xiao et al., 2024) in LMs using data from NLI datasets SST and MNLI (Socher et al., 2013; Williams et al., 2018a; Wang et al., 2019).

**Attention on sink tokens**. Fig. 1 illustrates the presence of attention sinks using attention maps, which show high attention scores are allocated to specific input tokens (i.e., sink tokens). The figure also shows sink tokens might receive similar (high) attention scores from all tokens. To empirically quantify these observations, we devise the measurements below.

Let $\mathbf{A} \in \mathbb{R}^{n \times n}$ denote an attention matrix over $n$ tokens for a single attention head. An element $a_{ij} \in \mathbf{A}$ denotes the attention on the $j$-th key token for the $i$-th query token. For the $i$-th (query) token, we have the following measurements :

$$\text{Average outer degree: } d_i = \sum_{k=1}^{n} a_{ki}/n,$$
$$\text{Attention deviation: } \Delta_i = \sqrt{\sum_{k=1}^{n} (a_{ki} - d_i)^2}/(nd_i) \tag{1}$$

We average $d_i$ and $\Delta_i$ across all attention-heads in each layer. $d_i$ is the averaged attention score allocated to the $i$-th token, and $\Delta_i$ is the per-degree attention score deviation to the $i$-th token's average outer degree. We study sink tokens with the largest average outer degrees, and calculate their attention deviations as sink attention deviations.

Fig.2 shows the layer-wise average outer degrees and sink attention deviations for BERT and RoBERTa models. In Fig.2 (a), tokens with top-3 largest outer degrees obtain 60% attention scores from input tokens, while the top-1 tokens obtain over 20% attention. This shows that a small number of tokens obtain major attention in self-attention layers. In Fig.2 (b), we observe that the sink attention deviations are mostly small, except for the first two layers. This shows that all tokens pay similar attention to the sink tokens in the top layers.

**Sink tokens are usually common tokens**. We also find that sink tokens in pre-trained LMs turn out to be *common tokens*, ones that appear in many language tasks. These common tokens include special tokens such as ('[SEP]'), punctuation ('.'), or the initial tokens in

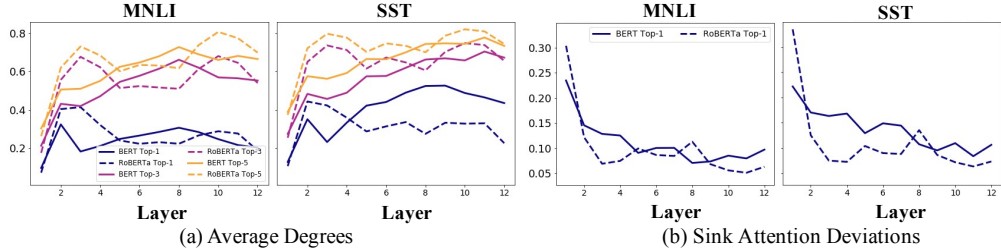

Figure 2: Average outer degrees and attention deviations on MNLI and SST data. (a) The cumulative average outer degrees of tokens with the top-1, top-3 and top-5 largest outer degrees. (b). The attention deviation of sink tokens with the top-1 largest outer degrees.

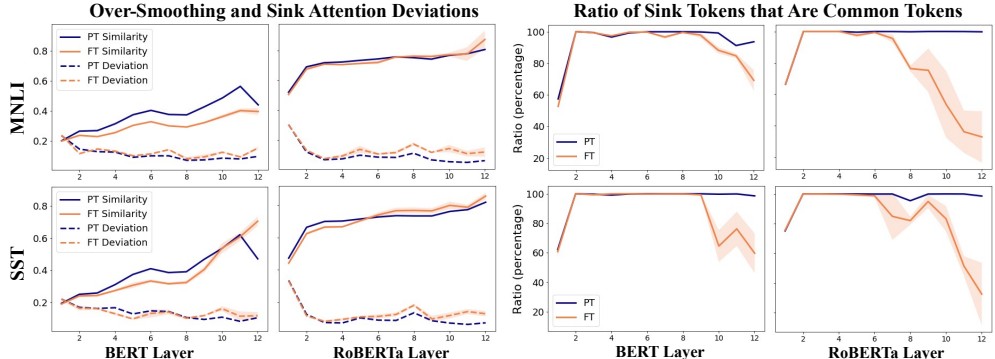

Figure 3: The left shows models' over-smoothing and attention deviation on sink tokens. The right shows the ratio of sink tokens that are common tokens across tasks. Here we consider special tokens, the punctuation '.' and the second token in the input as the common tokens. 'PT' stands for pre-training, and 'FT' stands for fine-tuning on 3k MNLI data.

inputs, which are not semantically significant. The right side of Fig. 3 shows the ratio of sink tokens with the top-1 largest outer degrees that are also common tokens at each layer. Almost all sink tokens are common tokens in the first several layers of the pre-trained model. Even after fine-tuning, the models still have high attention on them in the middle layers.

## 3.2 Connection between Over-Smoothing and Attention Sinks

We show the impact of attention sinks on single tasks by connecting them to an over-smoothing phenomenon, where the attention deviations on sink tokens are a key factor.

**Over-Smoothing in transformers**. Previous works have identified an over-smoothing phenomenon in transformer-based models: token representations become identical after several self-attention layers. Over-smoothing is closely related to models' task learning ability and their overfitting on a task (Shi et al., 2022). There are several factors that can lead to over-smoothing, and we focus on the effect of self-attention matrices here.

For a self-attention layer with an attention matrix $\mathbf{A} \in \mathbb{R}^{n \times n}$, let $\mathbf{H} \in \mathbb{R}^{n \times d}$ be its input token representations and $\mathbf{AH}$ its output. The over-smoothing problem is described as:

$$d_{\mathcal{M}}(\mathbf{AH}) \leq \sqrt{\lambda_{\max}} d_{\mathcal{M}}(\mathbf{H}).$$

The distance $d_{\mathcal{M}}(\mathbf{H}) := ||(\mathbf{I} - \mathbf{ee}^T)\mathbf{H}||_F$ measures the closeness between each token representation in $\mathbf{H}$ and the averaged token representation $\mathbf{ee}^T\mathbf{H}$, where $\mathbf{e} = n^{-\frac{1}{2}}[1, 1, ..., 1] \in \mathbb{R}^n$. $\lambda_{\max}$ is the largest eigenvalue of $\mathbf{A}^T(\mathbf{I} - \mathbf{ee}^T)\mathbf{A}$. When $\lambda_{\max}$ is small, representations after the attention layer will be closer to the averaged representations, which causes over-smoothing.

**Connection to attention sinks.** Over-smoothing has been identified in many transformer-based models. We analyze the eigenvalue $\lambda_{\max}$ to see the property of the attention matrix $\mathbf{A}$ under the over-smoothing circumstances (i.e., $\lambda_{\max}$ is small). With each attention score $a_{ij}$

and average outer degree $d_i$ defined in Section 3.1, $\lambda_{\max}$ is lower bounded as:

$$\lambda_{\max} \geq \max_i \sum_{k=1}^{n} \left( a_{ki} - d_i \right)^2. \tag{2}$$

The details are in Appendix A. When $\lambda_{\max}$ is small, the RHS of Eq. (2) has to be small. In particular, the $i$-th token which has the largest outer degree must have its deviation $\sum_{k=1}^{n}(a_{ki} - d_i)^2$ to be small. When $d_i$ is large (as shown in Fig. 2), to make the deviation small each attention $a_{ki}$ needs to be close to $d_i$. Therefore, all tokens have similar (high) attention on the token with the largest outer degree, which is an attention sink phenomenon.

We empirically show the connection between attention deviations $\Delta$ and over-smoothing in Fig. 3. The over-smoothing degree is reflected by the average cosine similarity between token representations (Shi et al., 2022). Going from lower to higher layers, we observe that the attention deviation *decreases* while the representation similarity *increases*. This validates the connection between attention sinks and the over-smoothing phenomenon.

**Impact of over-smoothing with attention sinks**. When over-smoothing occurs with attention sinks above, the sink token representations may dominate the averaged token representation, and make other token representations (including [CLS]) close to them. To learn a task, models may push sink token representations close to the task data representation (Fig. 4(a)). Since sink tokens may be irrelevant to tasks (Fig. 3 right), this may distort pre-trained features and make models less generalizable to OOD data (Kumar et al., 2022).

Comparing BERT and RoBERTa, we observe that pre-trained RoBERTa suffers more from over-smoothing (i.e., high representation similarity), corresponding with low attention deviations on sink tokens at the second and last several layers (Fig. 2(b)). Therefore, we hypothesize that RoBERTa may be more vulnerable to feature distortion in task learning, which is reflected by its distorted attention patterns (Fig. 1 and Fig. 3 right) after fine-tuning. This may also influence RoBERTa's CL capacity.

## 4 Attention Sink and Interference in Continual Learning

In this section, we first conduct a case study to show that attention sinks above can cause unnecessary interference when learning across tasks in CL. Then we discuss a transfer vs. interference trade-off induced by attention sinks, which inspires our method in Section 5.

### 4.1 Interference in Continual Learning

We study the following CL problem: A model continually learns a sequence of tasks, with no previous tasks' data accessible when learning new tasks, and no storage of previous tasks' model parameters. The model has different predictors for different tasks, while the encoder is shared. Each task $i$ consists of data $\mathcal{D}_i = (\mathbf{X}_i, y_i)$ where $\mathbf{X}_i$ is the input feature for task $i$, and $y_i \in \mathbb{R}$ is its target output.

When learning task $j$ after task $i$, one way to quantify the cross-task interference on the shared parameter $\theta$ is through the dot product between its (vectorized) gradients on the two tasks' losses (Riemer et al., 2019):

$$I(\theta; i, j) = \nabla_\theta L(\mathbf{X}_i, y_i) \cdot \nabla_\theta L(\mathbf{X}_j, y_j), \tag{3}$$

where $\cdot$ is the dot product operator on the flattened gradients. The interpretation is that interference that leads to forgetting happens when $I(\theta; i, j) < 0$, while the positive knowledge transfer may happen if $I(\theta; i, j) > 0$. For tasks that do not have knowledge transfer, the interference is expected to be 0.

### 4.2 Case Study: Attention Sink Can Cause Unnecessary Interference Between Tasks

We use a case study to show that attention sinks can propagate unexpected interference between irrelevant tasks. The study is based on the attention sink phenomenon characterized in Section 3, which showed: (1). models allocate high attention to sink tokens with small deviations; (2). sink tokens are usually common tokens shared across different tasks.

**Data.** We consider two *irrelevant* NLP tasks in a CL setting. For task $i$, we have data instance $(\mathbf{X}_i, y_i)$ where $\mathbf{X}_i$ consists of embeddings of input tokens. We make the following assumptions about tasks and data:

1. There is no knowledge transfer between the tasks and there should be no interference (positive or destructive) when learning one task after the other.

2. Assume there are $k$ common tokens (e.g., special tokens) in two tasks' data instances. For all other tokens in a task, we assume they are irrelevant to non-common tokens in the other task, with corresponding embeddings being orthogonal.

**Model.** We use a model consisting of two single-head self-attention layers. For each task $i$, the input $\mathbf{X}_i \in \mathbb{R}^{n_i \times d}$ consists of $d$-dimensional embeddings for $n_i$ tokens. Considering a regression problem (generalizable to classification), the prediction $\hat{y}_i$ is calculated as:

$$\hat{y}_i = \mathbf{b}_i^T (\mathbf{A}_i \mathbf{X}_i \mathbf{W}) \mathbf{v}_i,$$

where $\mathbf{A}_i \in \mathbb{R}^{n_i \times n_i}$ is the attention matrix in the first attention layer, $\mathbf{b}_i \in \mathbb{R}^{n_i}$ is the attention vector in the second attention layer that integrates all hidden representations for the target task prediction. Both $\mathbf{A}_i$ and $\mathbf{b}_i$ are obtained under the self-attention mechanism (Vaswani et al., 2017). $\mathbf{W} \in \mathbb{R}^{d \times d}$ is a transformation matrix. $\mathbf{v}_i \in \mathbb{R}^{d \times 1}$ is the predictor that maps the representation to an output value $\hat{y}_i$ for task $i$. The loss function is: $L(\hat{y}_i, y_i) = \mathbb{E}[\frac{1}{2}(\hat{y}_i - y_i)^2]$.

For simplicity, we sort $\mathbf{X}_i$ and $\mathbf{A}_i$ to make the $k$ common tokens have indices $\{1, ..., k\}$ and others have indices $\{(k+1), ..., n_i\}$. We assume the common tokens are sink tokens in $\mathbf{A}_i$.

**Claim.** *The interference on the transformation matrix $\mathbf{W}$ between task 1 and task 2 mainly depends on the outer degrees of sink tokens and the sink attention deviations.*

We calculate interference on the shared parameter $\mathbf{W}$ based on Eq. (3) and the model above:

$$I(\mathbf{W}) = (\hat{y}_1 - y_1)(\hat{y}_2 - y_2)(\mathbf{B}_1^T \mathbf{B}_2)(\mathbf{v}_1^T \mathbf{v}_2), \tag{4}$$

where $\mathbf{B}_1^T = \mathbf{b}_1^T \mathbf{A}_1 \mathbf{X}_1$ and $\mathbf{B}_2^T = \mathbf{b}_2^T \mathbf{A}_2 \mathbf{X}_2$. When both training losses are non-zero (which is the usual case in CL), interference in Eq. (4) depends on the correlation $\mathbf{v}_1^T \mathbf{v}_2$ between predictors and the correlation $\mathbf{B}_1^T \mathbf{B}_2$ between representations. Since the learned predictors may not be good enough to reflect the orthogonality between task 1 and task 2 (Kumar et al., 2022), we have to consider the interference caused by the correlation $\mathbf{B}_1^T \mathbf{B}_2$, discussed below.

*Step 1: decompose $\mathbf{B}_1$ and $\mathbf{B}_2$.* Generally, for any matrix $\mathbf{M}$, we denote $\mathbf{M}^{(ij)}$ as the $ij$-th element of $\mathbf{M}$. And for any vector $\mathbf{b}$, we denote $\mathbf{b}^{(i)}$ as the $i$-th element of $\mathbf{b}$.

Denote $\mathbf{d}_1$ as the vector of tokens' average outer degrees in attention matrices $\mathbf{A}_1$. For the attention $\mathbf{A}_1^{(ij)}$, we define its deviation to the $j$-th average outer degree as: $\epsilon_1^{(ij)} = \mathbf{A}_1^{(ij)} - \mathbf{d}_1^{(j)}$. Since the attention vector $\mathbf{b}_1$ is row-stochastic, we decompose $\mathbf{B}_1^T$ as:

$$\mathbf{B}_1^T = \underbrace{\sum_{i=1}^{n_1} \sum_{j=k+1}^{n_1} \mathbf{b}_1^{(i)} \mathbf{A}_1^{(ij)} \mathbf{X}_1^{(j\cdot)}}_{r_1: \text{ non-sink representations}} + \underbrace{\sum_{j=1}^{k} \mathbf{d}_1^{(j)} \mathbf{X}_1^{(j\cdot)}}_{s_1: \text{ sink representations}} + \underbrace{\sum_{i=1}^{n_1} \sum_{j=1}^{k} \mathbf{b}_1^{(i)} \epsilon_1^{(ij)} \mathbf{X}_1^{(j\cdot)}}_{\delta_1: \text{ representation deviations}}. \tag{5}$$

Similarly, we have $\mathbf{B}_2^T = r_2 + s_2 + \delta_2$, where $r_2$ denotes the non-sink representations, $s_2$ the sink representation and $\delta_2$ the sink representation deviations for task 2.

*Step 2: calculating $\mathbf{B}_1^T \mathbf{B}_2$.* Based on assumption 2 that non-sink token embeddings from two tasks are orthogonal, we have $r_1 r_2^T = 0$. Moreover, since sink tokens are common tokens that are supposed to be neutral to other tokens, we hypothesize that their embeddings are nearly orthogonal to other token embeddings (Appendix B). This makes $(s_1 + \delta_1) r_2^T + (s_2 + \delta_2) r_1^T \approx 0$. Then we have $\mathbf{B}_1^T \mathbf{B}_2 = (r_1 + s_1 + \delta_1)(r_2 + s_2 + \delta_2)^T \approx (s_1 + \delta_1)(s_2 + \delta_2)^T$.

Therefore, $\mathbf{B}_1^T \mathbf{B}_2$ depends largely on tasks' sink representations $s_1, s_2$ and their representation deviations $\delta_1, \delta_2$. Specifically, it is dominated by $s_1 s_2^T$ when: (1) each attention deviation $\epsilon^{(ij)}$

(a) Nearest neighbors of [CLS] token after fine-tuning

| | |
|---|---|
| **SNLI** (Neutral) | **S1**: Two women are embracing while holding to go packages. 
 **S2**: The sisters are hugging goodbye while holding to go packages after just eating lunch. |
| **BERT** | **L1**: [SEP] [SEP] . . The to to are are after while while go just go 
 **L6**: . . The are to women are packages sisters hugging while go after to em 
 **L12**: are The while hugging after just are . holding goodbye sisters eating [SEP] [SEP] . |
| **RoBERTa** | **L1**: women . <\s> <\s> . to The to are are go go holding holding after 
 **L6**: women <\s> . . Two <\s> The are packages packages sisters are embracing hugging go 
 **L12**: women <\s> Two <\s> The are after sisters are goodbye hugging while embracing |

(b) Models' capacities with and without sink tokens

| Data 
 Model | MNLI 
 (In-Task) | SNLI 
 (Transfer) | SNLI w/o Sink 
 (Transfer) |
|---|---|---|---|
| **BERT** | $69.2 \pm 0.2$ | $59.4 \pm 1.1$ | $46.4 \pm 1.6$ |
| - w/o Sink | $61.8 \pm 0.9$ | - | $50.9 \pm 4.9$ |
| **RoBERTa** | $76.1 \pm 0.7$ | $70.1 \pm 2.1$ | $39.3 \pm 1.4$ |
| - w/o Sink | $49.9 \pm 4.3$ | - | $45.8 \pm 1.6$ |

Figure 4: (a) After fine-tuning, sink tokens' representations can be close to data ([CLS]) representations, even though the sink tokens are irrelevant to the task. (b) Models trained on 3k MNLI data and evaluated on MNLI and SNLI data, with and without attention on common sink tokens. Sink tokens ensure models' capacity and their transfer to similar tasks.

in Eq. (5) is close to 0; or (2) the attention score $\mathbf{b}^{(i)}$ is close to 0 when the absolute deviation $|\epsilon^{(ij)}|$ is large, which makes $\mathbf{b}^{(i)}\epsilon^{(ij)}$ close to 0. When the sink tokens' outer degrees are large, the correlation $s_1 s_2{}^T$ can cause high interference between the orthogonal tasks 1 and 2.

### 4.3 Transfer vs. Interference

Since attention sinks on common tokens can propagate unnecessary interference, should we exclude sink tokens that are common tokens ('*common sink tokens*') when calculating attention in CL? To answer this question, we first train models with and without attention on common sink tokens, and then compare their in-task and transfer learning capacities. Results in Fig. 4(b) show that when discarding the attention on common sink tokens in training, models have a significant performance drop in the in-task evaluation.

In addition, common sink tokens may benefit tasks with positive knowledge transfer. In Fig. 4(b), we use models trained on MNLI data for zero-shot evaluation on SNLI data, which is for the same sentence entailment task but with a different data distribution. Results show that the models' transfer ability significantly drops after discarding the attention on common sink tokens. We hypothesize that this could be because sink tokens that are common across tasks can easily transfer knowledge on them.

The analysis above motivates us to balance the transfer and interference caused by attention sinks in task learning. Specifically, when allowing models to preserve relatively high attention on sink tokens, we in turn encourage them to pay attention to non-sink tokens that have relatively low attention on sink tokens. This may increase sink attention deviations, which help reduce interference and oversmoothing. We develop a pre-scaling model to achieve this goal, detailed in Section 5.

## 5 Method: Pre-Scaling For Diverse Attention

We introduce a pre-scaling mechanism that encourages models to allocate diverse attention to sink tokens and increase attention on non-sink tokens, when learning a downstream task.

As shown in Section 3.1, attention sinks exist in pre-trained models. However, since sink tokens are usually common tokens that are not semantically significant, their pre-trained representations may not contain much information related to downstream tasks. On the other hand, pre-trained representations of non-sink tokens may contain more information for task prediction. This motivates us to first allocate task-specific attention on tokens based on their pre-trained representations, for increasing attention on non-sink tokens.

We design a scaling layer to allocate attention scores on tokens based on their contributions to the task. To encourage diverse attention, we have different scaling of attention scores for different classes. The scaling layer is first learned through a probing stage to allocate high attention scores to non-sink tokens based on their pre-trained representations. Then we fine-tune the whole model.

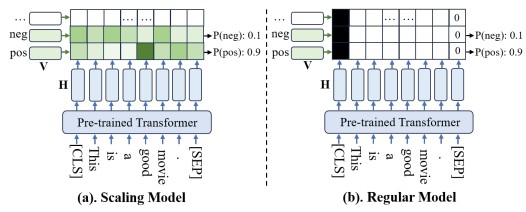

Figure 5: The scaling and regular model.

**Scaling layer.** For each task, let $\mathbf{H} = \{\mathbf{h}_1, ..., \mathbf{h}_n\}$ be the pre-trained representations of the $n$ input tokens, and $\mathbf{V} = \{\mathbf{v}_1, ..., \mathbf{v}_c\}$ the learnable class vectors for the $c$ classes in that task. Each token representation $\mathbf{h}$ and class vector $\mathbf{v}$ are $\mathbb{R}^d$ vectors. The scaling layer computes attention $\mathbf{A}_c$ on tokens for classes as:

$$\mathbf{A}_c = \text{softmax}\big(\mathbf{V}f(\mathbf{H})^T/\sqrt{d}\big), \tag{6}$$

where $f : \mathbb{R}^d \to \mathbb{R}^d$ is a learnable linear function. The output for class $i$ is calculated by:

$$p(i|\mathbf{H}) = \exp\big(\mathbf{A}_c^{(i\cdot)}\mathbf{H}\mathbf{v}_i\big)\big/ \sum_{i=1}^{c} \exp\big(\mathbf{A}_c^{(i\cdot)}\mathbf{H}\mathbf{v}_i\big),$$

where $\mathbf{A}_c^{(i\cdot)}$ is the $i$-th row of the attention $\mathbf{A}_c$, $\mathbf{v}_i$ is the $i$-th class vector in $\mathbf{V}$. We use the cross entropy loss to train the model with scaling layer.

**Two-Stage training.** For each task, we use a two-stage training process: (1). *probing*: the encoder is fixed and we only learn the scaling layer (including class vectors); (2). *fine-tuning*: the whole model, including the encoder and the scaling layer, is learned for the target task.

For sequential tasks in CL, we follow the task-incremental training where at each task, the loss is only computed over classes in that task. However, the scaling and prediction can be applied over classes in all tasks, and thus our model is general to the task-agnostic setting.

**Connection to probing then fine-tuning.** Our pre-scaling mechanism has connections to the probing-then-fine-tuning mechanism proposed in Kumar et al. (2022), since both use a similar two-step training process. However, our mechanism utilizes a scaling layer to gather diverse representations of all tokens instead of only using the representation of the [CLS] token for prediction. As shown in Fig. 5(b), probing-then-fine-tuning under the regular model is a special case in our mechanism while the attention scores in Eq. (6) are 1 for [CLS] token and 0 for other tokens. As claimed in Kumar et al. (2022), the two-stage training can reduce feature distortion by first learning good class vectors $\mathbf{v}$. These good class vectors may further benefit our pre-scaling mechanism in CL.

## 6 Experiments

### 6.1 Experimental Settings

**Datasets.** We evaluate four sequences of CL tasks: (1) **Yahoo Split**: a split of Yahoo dataset for news question-answer categorization (Zhang et al., 2015) with 5 disjoint tasks containing 2 classes each; (3) **DB**: a split of DBPedia data for Wikipedia article classification (Zhang et al., 2015) with 7 disjoint tasks containing 2 classes each; (4) **News Series**: a sequence of tasks on news-related data, including AG_news (news classification, 4 classes), MRPC (paraphrase detection, 2 classes) (Dolan & Brockett, 2005), RTE (text entailment, 2 classes) (Williams et al., 2018b) and SST (sentiment analysis, 2 classes) (Socher et al., 2013). For the above sequences, we randomly sample 1245 samples per class, which is the least number of class samples in our datasets.

**Baselines.** We consider two categories of baselines:

One category performs vanilla sequential learning for CL but has different training strategies on each single task, including (1) **FT**: a model where all parameters are sequentially updated; (2) **PT+FT** (Kumar et al., 2022): a model first trains the classifier in the probing stage and then fine-tunes the whole model; (3) **Prescale** (ours): a model first trains the classifier and a scaling layer in the probing stage and then fine-tunes the whole model (with scaling).

Another category is designed with specific CL techniques like experience replay, including (1) **ER**: a FT model storing all seen examples and performs sparse (1%) experience replay; (2) **A-GEM** (Chaudhry et al., 2019a): a FT model constraining on gradients to prevent degrading performance of previous tasks; (3) **MBPA++** (d'Autume et al., 2019): a FT model that stores and retrieves samples to locally adapt the model at inference time (Sprechmann et al., 2018). (4). **IDBR** (Huang et al., 2021b): a FT model with information-disentanglement-based regularization and replay. We also compare to IDBR without replay, denoted as **IDBR(-R)**; (5) **CTR** (Ke et al., 2021): an adapter-based task-incremental model with capsules

Table 1: Comparison between sequential models on CL. We report *ACC* and *FGT* with their standard deviations (*std*) on five random seeds. **Bold** scores are the best scores.

| Model | | Yahoo Split | | DB Split | | News Series | |
|---|---|---|---|---|---|---|---|
| | | $ACC_{std}$ | $FGT_{std}$ | $ACC_{std}$ | $FGT_{std}$ | $ACC_{std}$ | $FGT_{std}$ |
| **BERT** | Probing | $88.43_{0.06}$ | — | $99.30_{0.03}$ | — | $74.81_{0.46}$ | — |
| | FT | $86.19_{0.92}$ | $6.70_{1.08}$ | $66.22_{8.13}$ | $39.15_{9.47}$ | $68.98_{5.68}$ | $17.13_{7.48}$ |
| | PT+FT | $90.24_{0.53}$ | $2.23_{0.77}$ | $98.47_{2.23}$ | $1.64_{2.59}$ | $77.09_{2.11}$ | $8.16_{2.50}$ |
| | Prescale (ours) | $\mathbf{90.92}_{0.53}$ | $\mathbf{1.47}_{0.71}$ | $\mathbf{99.74}_{0.05}$ | $\mathbf{0.13}_{0.06}$ | $\mathbf{79.76}_{0.76}$ | $\mathbf{4.40}_{1.18}$ |
| **RoBERTa** | Probing | $88.06_{0.09}$ | — | $99.33_{0.01}$ | — | $68.27_{1.32}$ | — |
| | FT | $83.54_{4.66}$ | $10.91_{5.74}$ | $71.94_{7.48}$ | $32.53_{8.73}$ | $70.61_{4.42}$ | $18.24_{5.21}$ |
| | PT+FT | $90.76_{0.86}$ | $2.14_{1.06}$ | $99.68_{0.24}$ | $0.21_{0.29}$ | $79.39_{2.00}$ | $8.01_{3.24}$ |
| | Prescale (ours) | $\mathbf{90.92}_{0.77}$ | $\mathbf{1.95}_{1.01}$ | $\mathbf{99.78}_{0.08}$ | $\mathbf{0.09}_{0.11}$ | $\mathbf{81.59}_{1.74}$ | $\mathbf{4.16}_{2.33}$ |

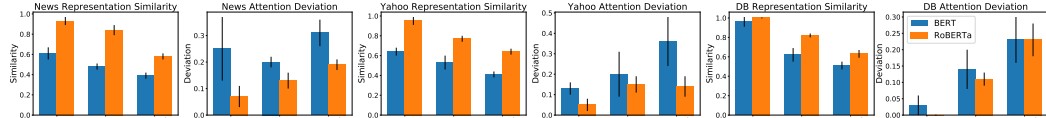

Figure 6: The last layer's representation similarity (for over-smoothing) and attention deviations (for attention sinks) for BERT and RoBERTa models on CL tasks. For an overall view, we average attention deviations over the tokens with top-5 largest outer degrees.

and task transfer routing; (6) **L2P** (Wang et al., 2022): a prompt-based model that learns to dynamically prompt for different data and tasks. We also compare models under multi-task learning (**MTL**) and separate learning for each task (**Separate**) as non-CL baselines to show the performance gap from CL models to them. Detailed settings are in Appendix C.

We compare BERT-base and RoBERTa-base models in sequential learning, and use BERT-base for CL-specific models. For BERT models, we use learning rate 2e-5 to train 3 epochs for each task; and for RoBERTa models, we use learning rate 1e-5 to train 5 epochs per task. For probing and pre-scaling, we use the learning rate 5e-4 to train the classifier.

**Metrics.** We train models in a task-incremental setting where task identifiers are given (task-aware). We evaluate models' CL performance by evaluating their average accuracy (*ACC*) and forgetting (*FGT*) on the sequence (Chaudhry et al., 2019b), with or without task identifiers (task-agnostic). For analysis, we evaluate models' over-smoothing and attention deviations on sink tokens using metrics in Eq. (1).

## 6.2 Results

**RoBERTa does not always outperform BERT in CL.** Table 1 compares BERT and RoBERTa's CL performance under sequential learning. With fine-tuning, RoBERTa does not always outperform BERT in CL despite its high pre-trained capacity. Specifically, RoBERTa has a lower accuracy than BERT on Yahoo Split and a higher forgetting on News Series. This may relate to its higher over-smoothing and lower sink attention deviations than BERT, which make it more vulnerable to feature distortion and easier to propagate interference.

**Prescaling improves RoBERTa's CL capacity.** With PT+FT, RoBERTa consistently outperforms BERT on CL tasks. We believe that is because PT+FT first learns good class vectors **v**, which reduces feature distortion in each single task and then benefits CL. After applying our prescaling method, BERT and RoBERTa achieve further improvements in CL tasks.

**Prescaling increases attention deviations.** In Fig. 6, we compare models' representational similarity and attention deviations after CL. After prescaling, RoBERTa's representation similarity decreases while the attention deviations increase. This suggests that our prescaling can encourage diverse attention, which reduces over-smoothing and benefits CL.

**Prescaling model outperforms CL models with replay.** In Fig. 5, we compare our prescaling model to CL-specific models to evaluate its overall CL capacity. Without experience replay or progressively storing model parameters, Prescale achieves overall best accuracies on CL tasks. Even for the News Series sequence which has knowledge transfer between tasks, Prescale outperforms replay-based models that are effective in this scenario. This validates the effectiveness of our prescaling mechanism in CL.

Table 2: Results for task-incremental learning. We report *ACC* and *FGT* with their *std* on five random seeds. **Bold** scores are the best scores and underline scores are the second best.

| | Model | Yahoo Split | | DB Split | | News Series | |
|---|---|---|---|---|---|---|---|
| | | *ACC std* | *FGT std* | *ACC std* | *FGT std* | *ACC std* | *FGT std* |
| **CL** | ER | 87.42 0.52 | 5.61 0.68 | 91.05 10.20 | 10.20 10.14 | 75.47 3.93 | 7.81 5.27 |
| | A-GEM | 89.43 0.58 | 2.95 0.64 | 94.71 4.70 | 5.98 5.49 | 75.90 3.34 | 6.60 3.84 |
| | MBPA++ | 86.50 2.78 | 6.62 2.82 | 97.17 3.76 | 3.09 3.68 | 72.55 5.50 | 9.64 3.99 |
| | IDBR (-R) | 89.32 1.46 | 2.74 1.35 | 96.47 4.67 | 3.95 4.66 | 72.36 2.93 | 8.67 4.23 |
| | IDBR | 90.48 0.55 | 1.32 0.64 | **99.84** 0.03 | 0.04 0.03 | 76.90 1.98 | 3.24 2.50 |
| | CTR | 87.06 1.23 | 1.28 0.93 | 99.04 0.95 | 0.29 0.35 | 75.12 3.09 | 3.40 2.92 |
| | L2P | 90.82 0.58 | 0.60 0.56 | 99.63 0.36 | 0.29 0.36 | 73.99 2.36 | 3.43 2.42 |
| **Sequential** | FT | 86.19 0.92 | 6.70 1.08 | 66.22 8.13 | 39.15 9.47 | 68.98 5.68 | 17.13 7.48 |
| | Prescale (ours) | **90.92** 0.53 | 1.47 0.71 | 99.74 0.05 | 0.13 0.06 | **79.76** 0.76 | 4.40 1.18 |
| **Non-CL** | Separate | 92.25 0.04 | — | 99.87 0.01 | — | 83.72 0.53 | — |
| | MTL | 92.27 0.05 | — | 99.88 0.01 | — | 82.04 0.90 | — |

Table 3: *ACC* and *FGT* of different scaling strategies on News Series.

Table 4: *ACC* on task-agnostic evaluations for DB and Yahoo Split.

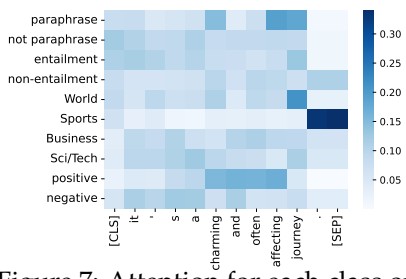

Figure 7: Attention for each class on tokens in SST (from News Series).

| | Model | *ACC* | *FGT* |
|---|---|---|---|
| **BERT Prescale** | Uniform | 78.98 | 5.32 |
| | Sink | 76.08 | 9.27 |
| | Full | 79.76 | 4.40 |
| **RoBERTa Prescale** | Uniform | 79.44 | 7.07 |
| | Sink | 79.09 | 9.26 |
| | Full | 81.59 | 4.16 |

| | Model | *DB* | *Yahoo* |
|---|---|---|---|
| **BERT** | FT | 15.90 | 36.19 |
| | PT+FT | 72.41 | 53.34 |
| | Prescale | 70.38 | 53.21 |
| **RoBERTa** | FT | 18.71 | 36.24 |
| | PT+FT | 67.32 | 52.98 |
| | Prescale | 77.55 | 53.51 |

### 6.3 Ablation Study

**Scaling strategies.** In Table 3, we compare our prescaling strategy (Full) to two other scaling strategies: one uniformly distributing attention to all tokens (Uniform); another learning to scale attention only on common sink tokens including special tokens, the punctuation '.' and the second token in the sentence (Sink). Results show that only scaling attention on common sink tokens does not yield improvements to PT+FT, and uniform scaling does not give as much improvement as full scaling. These suggest that the effectiveness of our prescaling strategy does not come only from having distributed attention on tokens.

**Scaling visualization.** Fig. 7 shows a heapmap of the scaled attention on the SST data (with task classes {positive, negative}) after training the model on News Series. For the corresponding positive/negative classes, we observe that attention is also distributed on task-related tokens (e.g., charming, affecting) besides common tokens.

**Task-agnostic evaluation.** In Table 4, we evaluate models in the task-agnostic setting after task-aware training to show the separation of data representations across tasks. Both PT+FT and Prescale perform better than FT. This may relate to their larger attention deviations and less over-smoothing (Fig. 6), which make data representations contain more information from non-sink tokens with different distributions across tasks. On BERT, PT+FT outperforms Prescale. We hypothesize that this is because BERT is pre-trained with the next sentence prediction, and thus [CLS] may contain more general sentence-level information across tasks than the learned scaling. On the other hand, for RoBERTa which does not have sentence-level pre-training, Prescale performs better than PT+FT.

## 7 Conclusion

In this paper, we study an attention sink phenomenon that can cause pre-trained models to perform inferior in CL tasks, despite their pre-trained capacity on downstream tasks. Specifically, we find that the small attention deviation on sink tokens and the fact that sink tokens are usually common tokens across tasks can cause the model having an over-smoothing problem and easily propagate interference during CL. To mitigate such interference, we propose a prescaling mechanism which first learns a scaling layer during probing to allocate diverse attention on non-sink tokens, and then fine-tunes the model with scaling. Results show that our pre-scaling method outperform most CL models and other scaling strategies.

**Acknowledgments**

We thank the anonymous reviewers for their helpful feedback to improve the earlier draft of this paper. This material is based on research supported in part by the Air Force Research Laboratory (AFRL), DARPA, for the KAIROS program under agreement number FA8750-19-2-1003.

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

# A  Connection between Over-Smoothing and Attention Sinks

Denote $\mathbf{P} = \mathbf{A}^T(\mathbf{I} - \mathbf{e}\mathbf{e}^T)\mathbf{A}$ and $\lambda_{\max}$ is the largest eigenvalue of $\mathbf{P}$. Denote the layer's attention matrix as $\mathbf{A} \in \mathbb{R}^{n \times n}$ ($n$ is the number of input tokens) and its $ij$-th element as $a_{ij}$. The eigenvalue $\lambda_{\max}$ is lower bounded by:

$$\lambda_{\max} \geq \max_i \sum_{k=1}^{n} \left(a_{ki} - d_i\right)^2,$$

where $d_i = \sum_{k=1}^{n} a_{ki}/n$ is the average outer degree of the $i$-th token.

*Derivation.* Each element in $\mathbf{P}$ can be written as:

$$\mathbf{P}_{ij} = \sum_{k=1}^{n} \left[a_{ki}a_{kj} - \left(\frac{1}{n}\sum_{q=1}^{n} a_{qi}\right)\left(\frac{1}{n}\sum_{q=1}^{n} a_{qj}\right)\right],$$

where $a_{ij}$ represents the $ij$-th element in $\mathbf{A}$. Because $\mathbf{P}$ is symmetric and positive semi definite, its max eigenvalue $\lambda_{\max}$ is real and positive, which also satisfies:

$$\lambda_{\max} = \max_{||\mathbf{x}||=1} \mathbf{x}^T \mathbf{P} \mathbf{x},$$

where $\mathbf{x} \in \mathbb{R}^n$. Set $\{\mathbf{x}\} = \{\mathbf{x}_1, ..., \mathbf{x}_n\}$ as a set of unit vectors, where each $\mathbf{x}_i$ has the element as 1 at the $i$-th place and others as 0. Then $\lambda_{\max}$ is lower bounded as:

$$\lambda_{\max} \geq \max_i \sum_{k=1}^{n} \left(a_{ki} - d_i\right)\left(a_{ki} + d_i\right)$$

$$= \max_i \sum_{k=1}^{n} \left(a_{ki} - d_i\right)^2 + 2d_i \underbrace{\sum_{k=1}^{n} \left(a_{ki} - d_i\right)}_{= \, 0 \text{ by the definition of } d_i}$$

$$= \max_i \sum_{k=1}^{n} \left(a_{ki} - d_i\right)^2$$

The RHS above can be further decomposed as $\max_i d_i^2 \sum_{k=1}^{n} \left(a_{ki} - d_i\right)^2 / d_i^2$ to reflect the effects of the outer degree and the per-degree attention deviations.

There are several cases that can make $\lambda_{\max}$ small. When the largest average degree $d_i$ is large, $\lambda_{\max}$ is small when its per-degree attention deviations are small. On the other hand, when the largest average degree is small, $\lambda_{\max}$ can be small even when the per-degree attention deviations are relatively large. In the paper, we focus on the case when the largest average is large, for the observed attention sink phenomenon.

# B  Correlation between Sink and Non-Sink Token Embeddings

In Section 4, the interference $\mathbf{B}_1^T \mathbf{B}_2$ depends on correlations between representations of the sink and non-sink tokens, which is related to correlations between their embeddings in $\mathbf{X}_1$ and $\mathbf{X}_2$. Here we empirically calculate the correlations (i.e., dot product) between embeddings of the sink and non-sink tokens in BERT and RoBERTa-base, to verify our hypothesis that (common) sink tokens' embeddings are nearly orthogonal to other tokens' embeddings. Results are shown in Fig. 8.

For both BERT and RoBERTa, embeddings of sink tokens (e.g, [CLS] and [SEP]) have close to 0 correlations to most other token embeddings. On BERT, the punctuation '.' has negative correlations to many other tokens, while on RoBERTa the distribution of its correlations is also centered around 0. We also randomly sample non-sink tokens from the vocabulary and show their embeddings' correlation distributions as a reference. On BERT, the embedding of non-sink token 'exchange' tends to have positive (non-zero) correlation to other tokens' embeddings. On RoBERTa, although the embedding of the non-sink token 'aution' has centered to 0 correlations to other tokens' embeddings, it has large correlations up to 8. However, the correlation distributions of cls () and sep (<\s>) tokens' embeddings have a much smaller range.

Compared to the correlation to other tokens' embeddings, we also compute the self-correlation on sink tokens' embeddings as shown in Fig. 9. The embeddings' self-correlations can not be ignored on both BERT and RoBERTa. When the common sink

tokens are allocated high attention, the correlation between sink token representations from two tasks may be large, leading to interference in CL.

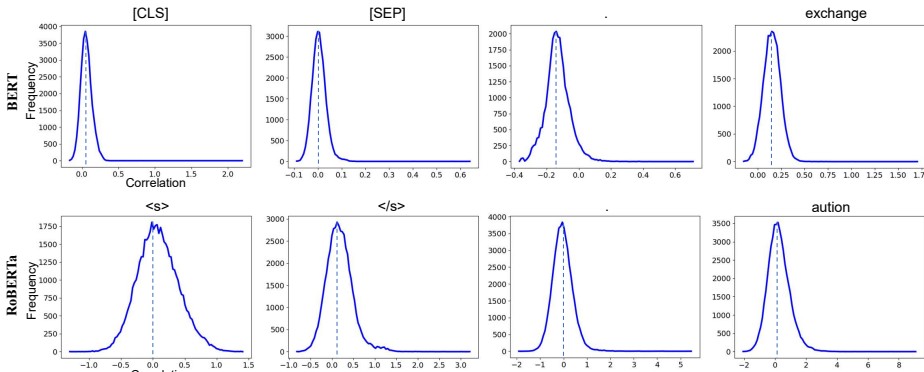

Figure 8: Correlation distributions between pre-trained sink token embeddings and all other tokens' embeddings based on BERT and RoBERTa. The right-most column shows the correlation distribution of a randomly selected token embedding for reference.

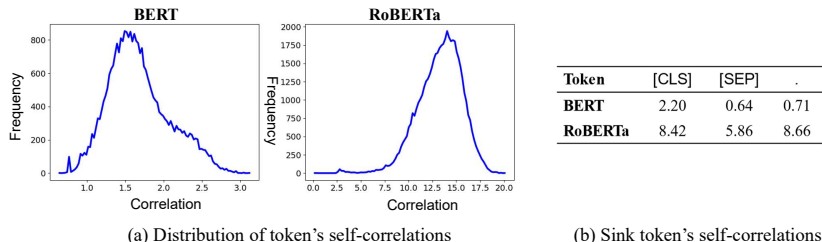

(a) Distribution of token's self-correlations   (b) Sink token's self-correlations

Figure 9: (a). Distribution of token embeddings' self-correlations on BERT and RoBERTa. (b). Self-correlations of sink tokens' embeddings.

## C   Detailed Experimental Settings

In Section 5, we train all models with task-incremental settings (i.e., training with the loss only on classes in that task), while evaluating them in both task-incremental and class-incremental settings. We perform all experiments on one Nvidia RTX A6000 machine.

We provide detailed experimental settings of baselines below:

- **Probing**: We fix the encoder and only train the classifier. We train 5 epochs for each task in BERT and RoBERTa, with the learning rate 5e-4.

- **FT**: We fine-tune the whole model, including the encoder and classifier. We train 3 epochs for each task in BERT, with the learning rate 2e-5; and train 5 epochs for each task in RoBERTa, with the learning rate 1e-5.

- **PT+FT**: We first train the classifier with the same setting in Probing, and then train the whole model with the same setting in FT.

- **Prescale**: We first train the classifier and the scaling layer with the learning rate 5e-4 for 5 epochs, and then train the whole model with the same setting in FT.

- **IDBR**: We train IDBR with the learning rate 3e-5 for 3 epoches per task. We follow the k-means memory selection rule, and the replay batch size is 16 (training batch size) × number of tasks in the memory.

- **CTR**: We follow the settings in the original paper, training 5 epochs for each task.

- **L2P**: We have the prompt pool with 100 prompt tokens and select 50 of them to prepend to the input. We train the model with the learning rate 1e-3 for 20 epochs for each task.

- **ER**: We apply sparse experience replay with 1% replay ratio. At each replay time, we sample 32 samples from the memory and perform one-step gradient descent based on them.

- **A-GEM**: We store all previous data in the memory. At each gradient step, we randomly extract 32 samples from the memory and apply the A-GEM gradient projection.

- **MBPA++**: We fine-tune the model with ER and then adapt the model at the inference time. At the inference time, we retrieve 32 nearest samples in the memory for local adaptation.

