# OpenReview forum: "Does RoBERTa Perform Better than BERT in Continual Learning: An Attention Sink Perspective"
_colmweb.org/COLM/2024/Conference — COLM_

### Official Review · Reviewer_ASC9 · 2024-05-10

**Rating:** 6
**Confidence:** 4
**Ethics Flag:** 1

**Summary:**

This paper discusses continual learning from the perspective of "attention sink." It initially notes that attention often concentrates on specific tokens for a single task. The authors suggest that this could cause interference in continual learning, as the sink tokens might hold information specific to one task. Consequently, they introduce a pre-scaling mechanism to promote diverse attention scores on sink tokens.

**Questions To Authors:**

See above

**Reasons To Accept:**

1. I believe the idea is logical. Interference is the primary reason for forgetting. Anything specific to a single task may cause this interference, including attention sink tokens.

2. There are both empirical and theoretical analyses.

**Reasons To Reject:**

1. This paper focuses on task-incremental learning. As noted in [1], a significant non-CL baseline in TIL is to train each task separately. This can help determine whether the proposed methods effectively balance transfer and interference.

2. The BERT and RoBERTa models are quite outdated. I believe the community would be more interested in whether the proposed method remains effective with the latest LLMs

3. Given that the pre-trained model has been updated, how does the current method prevent forgetting the original pre-trained knowledge?

[1]: Continual Learning of Natural Language Processing Tasks: A Survey, Ke and Liu, arXiv 2023

---

> ### Author Rebuttal · Authors · 2024-05-31
>
> **1. Non-CL baseline**
>
> We list the results of training each task separately below.
> | Model | Yahoo Split | DB Split |News Series|
> |----------|----------|----------|----------|
> |  FT-BERT  |   92.25±0.04  | 99.87±0.01  | 83.72±0.53|
> |  FT-RoBERTa |   92.43±0.07 |  99.83±0.01  |85.01±1.54|
>
>  We have also provided the MTL performance in the paper.
>
> **2. BERT and RoBERTa models are quite outdated**
>
> BERT and RoBERTa are still widely used in CL, especially for foundational analyses [1, 2]. We believe our analyses and methods are generalizable to larger models. First, our pre-scaling layer is applied after the LM encoder, which can be easily adapted to different LMs. Second, our analysis is general to different LMs. As long as there are attention sinks on common tokens (e.g. as has been observed in new-generation LLMs [3]), our analysis of interference based on them is still valid.
>
> We show the (pre-trained) attention matrices on LLama 3 8B in our classification tasks in the [link]( https://drive.google.com/file/d/10QbuAjtZ5qJQqD1Ve0ATCooF3yZq-5qx/view?usp=drive_link) (Figure B). The results show that LLama 3 also has attention sinks on the first token (common token), thus our analysis is still valid. However, for the latest LLMs, we may need new tasks and benchmarks for CL evaluation, which are not well-explored in previous works, and we keep this in future works.
>
> **3. Prevent forgetting the original pre-trained knowledge**
>
> The current method preserves the pre-trained knowledge in the following ways:
>
> - The additional probing stage is proven to reduce the distortion of pre-trained representations [3], which in turn helps to preserve the pre-trained knowledge.
>
> - In our model, the pre-trained knowledge is used to pre-scale the weights of each token in the data. This helps to use the pre-trained knowledge for the current task’s learning, and may further reduce the distortion of pre-trained representations. As shown in the paper Figure 1, compared to fine-tuning, pre-scaling better preserves the attention patterns in the pre-trained model.
>
> [1]. An Empirical Investigation of the Role of Pre-training in Lifelong Learning. Mehta et.al, JMLR 2023
>
> [2]. Can BERT Refrain from Forgetting on Sequential Tasks? A Probing Study. Tao et.al, ICLR 2023
>
> [3]. Efficient Streaming Language Models with Attention Sinks. Xiao et. al, ICLR 2024
>
> [4]. Fine-Tuning can Distort Pretrained Features and Underperform Out-of-Distribution. Kumar et. al, ICLR 2022

---

### Official Review · Reviewer_MZAH · 2024-05-12

**Rating:** 6
**Confidence:** 3
**Ethics Flag:** 1

**Summary:**

The paper presents a novel perspective on the performance of pre-trained language models (LMs) in continual learning (CL). The authors observe that despite higher single-task performance, RoBERTa does not consistently outperform BERT in CL settings. The paper introduces the concept of "attention sink" and its connection to over-smoothing, which may lead to interference in learning subsequent tasks. To address this, the authors propose a pre-scaling mechanism to encourage attention diversity and reduce interference.

**Questions To Authors:**

1. Is the analysis of single-head self-attention in Section 4.2 not without loss of generality for multi-head self-attention?
2. Is the phenomenon that Sink Tokens are Usually Common Tokens in Section 3.2 beneficial or harmful to CL? If harmful, then Fig. 2 right indicates that the proportion of Sink Tokens that are Common Tokens decreases after RoBERTa fine-tuning, which should mitigate the damage to CL.
3. Is the problem of RoBERTa in CL mentioned in the introduction caused by a lack of stability or plasticity? That is, is the performance low on new tasks or old tasks, or both? Specific experimental results analysis is hoped for.

**Reasons To Accept:**

1. The paper presents a unique analysis of the attention sink phenomenon and its impact on CL, which is a novel contribution to the field of NLP.
2. The proposed pre-scaling mechanism is theoretically sound and is supported by empirical evidence through extensive experiments on various CL tasks.
3. The paper demonstrates improvements in CL performance without relying on experience replay or parameter storage from previous tasks, which is a substantial advancement in the efficiency of CL models.

**Reasons To Reject:**

1. The performance advantage is not very pronounced, and it is hoped that verification can be done on more challenging tasks. Since most of the experiments in the paper are binary classification tasks, the absolute values of accuracy are already very high, making the differences between various comparison methods not significant enough.
2. Does the two-step training process significantly increase computational costs?
3. Is it reasonable to assume that tokens other than common tokens are definitely irrelevant in the unrelated tasks in Section 4.2?
4. Is it reasonable to assume that unrelated tasks in the continual learning tasks are irrelevant? If tasks are completely unrelated, then directly learning separate models for each task would be the best solution. Therefore, the trade-off between Transfer and Interference analyzed in Section 4.3 is more important.

---

> ### Author Rebuttal · Authors · 2024-05-31
>
> **1. More experiments**
>
> We show more experiments on (1). *Review* [1] including tasks with more classes; (2). *GLUE* including five GLUE tasks (more challenging), and two review tasks (more classes).
> |Model|Review Acc|Review FGT|GLUE Acc|GLUE FGT|
> |-|-|-|-|-|
> |FT|69.01±3.04|6.90±3.80|67.06±2.15|12.76±2.37|
> |IDBR|71.26±0.88|1.97±1.10|70.57±1.42|4.84±1.83|
> |Prescale|**72.56±0.66**|2.33±0.86|**72.20±1.55**|7.68±2.00|
> |FT-RoBERTa|68.90±3.33|8.92±4.16|65.07±3.60|18.20±4.26|
> |Prescale-RoBERTa|**74.11±0.48**|1.92±0.62|**74.45±1.54**|7.17±1.64|
>
> Our models consistently outperform baseline models on BERT and achieve further improvements on RoBERTa.
>
> **2. Computation**
>
> The probing stage takes less than 40% training time and 30% GPU memory compared to full fine-tuning.
>
> **3. Assumptions in Sec 4.2**
>
> We make assumptions in Sec 4.2 to show the effect of attention sinks in propagating interference. However, our Eq. (5) is generalizable to relevant tasks and the case where non-common tokens are relevant. In these cases, the interference can be caused by both common and other tokens, which depend on correlations between data of specific tasks and are hard to identify. Since we have observed the attention sinks on common tokens, we focus on their effects in the analysis. In experiments, we also test on relevant tasks (in News Series) with other related tokens.
>
> For comparison to isolation-based models, please see Point 4 of the response to Reviewer JZWa.
>
>
> **4. Multi-head attention (MHA)**
>
> For each head, our analysis holds and after concatenation for MHA, attention sinks will cause interference at corresponding dimensions.
>
> **5. Effect of common sink tokens**
>
> The effect depends on the correlation between tasks. It may be harmful if sink tokens propagate unexpected interference between specific tasks. However, as in Sec 4.3, sink tokens may also make knowledge transfer easier.
>
> As shown in the paper Figure 1, although the common sink tokens decrease after fine-tuning, the attention patterns deviate far from the pre-trained patterns, which may not well preserve the pretrained knowledge, causing the model to lose generalization and forget.
>
> **6. Stability vs. plasticity**
>
> In our experiments, fine-tuning RoBERTa has higher averaged accuracy but more forgetting than BERT on News Series, which indicates it performs better on new tasks while forgetting more (lack of stability).
>
> [1]. Episodic Memory in Lifelong Language Learning. d'Autume et.al, Neurips 2019.

---

### Official Review · Reviewer_Gejt · 2024-05-12

**Rating:** 6
**Confidence:** 3
**Ethics Flag:** 1

**Summary:**

The paper studies the attention patterns of BERT and ROBERTA and they evaluate if an over-smoothing pattern in the token representations is influencing the learning patterns in a continual learning setup. Furthermore, they propose a mechanism for encouraging the models to have a more diverse attention behaviour. This model change improves the adaptation capabilities of the models to new tasks int a continual learning framework.

**Reasons To Accept:**

Firstly, the authors follow a principled approach to study the phenomenon and the issues it causes on the model behaviour and then they propose a sound and clean solution which also has a significant positive impact on model performance.

They also have a strong experimental design and framework that should be easily adaptable to other models. Also, the models are throughly evaluated and the results are interpreted in a clear manner.

**Reasons To Reject:**

I think that the paper requires a more comprehensive perspective on the impact of the studied phenomenon in the tasks that BERT and Roberta are generally used.

Second, I would like to see a better coverage of how this work can be extended to other models and tasks and more specifically on the new class of language models that would have a bigger capacity of encoding the information from multiple tasks. It would also help to provide resources that would facilitate applying the proposed techniques and evaluations to other models.

---

> ### Author Rebuttal · Authors · 2024-05-31
>
> **1. A more comprehensive perspective on the impact of attention sinks**
>
> Thanks for the suggestion. BERT and RoBERTa are generally fine-tuned for language understanding tasks, like those in the GLUE benchmark. We show BERT and RoBERTa’s attention deviation (related to attention sinks) and representational similarity (to measure the over-smoothing degree) on three more GLUE tasks in this [link]( https://drive.google.com/file/d/10QbuAjtZ5qJQqD1Ve0ATCooF3yZq-5qx/view?usp=drive_link) (Figure A). Results have a similar trend as shown in the main paper Figure 2.
> &nbsp;
>
> On single tasks, we observe that if a model allocates too much attention to high-degree tokens (attention sinks), it may result in the over-smoothing phenomenon (reflected by representation similarity), which decreases the model’s single-task learning capacity [1]. When the attention sinks are on task-irrelevant tokens (e.g., common tokens including punctuations), there may be spurious correlations based on them during tuning, which degrade the model’s generalization ability [2].
>
> **2. How to apply the proposed method and evaluations to the latest LLMs**
>
> - Our method and evaluation in this paper are generalizable to other transformer-based models. First, our pre-scaling layer is applied after the LM encoder, which can be easily adapted to different LMs. Second, our evaluation of attention sinks (attention deviations) are applicable to attention matrices of different LMs. As long as there are attention sinks on common tokens (e.g. as has been observed in new-generation LLMs [3]), our analysis of interference based on them is still valid.
>
> - We show the (pre-trained) attention matrices on LLama 3 8B of our classification tasks in this [link]( https://drive.google.com/file/d/10QbuAjtZ5qJQqD1Ve0ATCooF3yZq-5qx/view?usp=drive_link) (Figure B). The results show that LLama 3 also has attention sinks on the first token (common token), thus our analysis is still valid. However, for the latest (more capable) LLMs, we may need new tasks and benchmarks for CL evaluation, which are not well-explored in previous works. We will keep this in our future works.
>
> [1]. Revisiting Over-smoothing in BERT from the Perspective of Graph. Shi et.al, ICLR 2022
>
> [2]. Are All Spurious Features in Natural Language Alike? An Analysis through a Causal Lens. Joshi et al, EMNLP 2022.
>
> [3]. Efficient Streaming Language Models with Attention Sinks. Xiao et. al, ICLR 2024

---

### Official Review · Reviewer_JZWa · 2024-05-12

**Rating:** 6
**Confidence:** 4
**Ethics Flag:** 1

**Summary:**

The paper studies the attention sink and its effect on continual learning. It also compared BERT and RoBERTa for continual learning with attention sink.

**Questions To Authors:**

See the weaknesses section

**Reasons To Accept:**

The paper studied continual learning from a new angle. Previous research on CL has not dealt with attention sink.

The results are ok although there is no major improvement.

**Reasons To Reject:**

It is sometimes confusing whether you are discussing class-incremental learning (CIL) or task-incremental learning (TIL) when you talk about the attention sink's impact. It is only clear in the experiment section that your work is about TIL. Please make the CL setting clear earlier.

In TIL, if you use a parameter isolation-based method, there is not much impact. See [1]

TIL is very much a solved problem. Its results mainly depend on individual task performance and knowledge transfer. It is unclear if the state-of-the-art TIL method is used, we can see any noticeable impact.

What is FGT? Is it the forgetting rate? For TIL, there should not be forgetting. See  [1,2]

[1] "Sub-network Discovery and Soft-masking for Continual Learning of Mixed Tasks." EMNLP-2023;
[2] "Beyond not-forgetting: Continual learning with backward knowledge transfer." NeurIPS-2022.

---

> ### Author Rebuttal · Authors · 2024-05-31
>
> **1. Make the CL setting clear earlier**
>
> Thanks for your suggestions. We will add a clarification of the CL settings earlier in the paper.
>
> **2. What is FGT?**
>
> Sorry about the confusion. FGT is the forgetting rate defined in [3].
>
> **3. TIL is very much a solved problem and there should not be forgetting**
>
> First, we would like to clarify that our work focuses on the influence of an attention-sink phenomenon on CL, a characteristic of LM and its tuning, but not on the CL strategy outside the LM. Therefore, we study the CL model which does not have (*a*) experience replay and (*b*) storage of previous tasks’  parameters. Under the above restrictions, models may still forget in TIL, supported by recent CL works like [4]. We originally claimed this in Section 4.1 but it might not be clear enough. Thanks for pointing it out and we will further clarify this.
>
> Both methods in reference [1, 2] (from the review) are based on stored task models (masks) or task gradients, which do not satisfy our restriction above (and our method can be combined with them). In addition, we notice that forgetting is also observed for [2]’s method in [1]’s experiments, which may suggest that it is still an issue in TIL, even if we store the information of previous tasks.
>
> **4. Comparison to isolation-based method**
>
> Although isolation-based methods can have no forgetting in TIL, at inference time, they require specification of the task to select which isolated part to use. Therefore, they can not be directly applied to CIL for inference. However, our method focuses on a characteristic (attention sink) of LM and can be applied to CIL evaluation after the TIL training. And it can achieve improved CIL performance as well. The results are shown below.
>
> | Model | DB Split (CIL Eval) | Yahoo Split (CIL Eval) |
> |-|----------|----------|
> |  FT  |   15.90 ± 3.35  |   36.19 ± 4.81  |
> |  IDBR (w/ replay)  |   62.07 ± 12.68  |   40.20 ± 4.61  |
> |  Prescale (ours)  |   **70.38 ± 4.84**  |   **53.21 ± 4.46**  |
> |  FT-RoBERTa  |   18.71 ± 3.29  |   36.24 ± 4.37  |
> |  Prescale-RoBERTa (ours)  |**77.55 ± 10.99**|**53.51 ± 2.63**|
>
> These results further support our claim that increasing attention diversity when tuning LM can help CL (for both CIL and TIL).
>
> [3] Continual learning with tiny episodic memories. Chaudhry et.al,  ICML WS 2019
>
> [4] An Empirical Investigation of the Role of Pre-training in Lifelong Learning. Mehta et.al, JMLR 2023

---

### Author Response · Authors · 2024-06-04
**Gentle nudge on our author response.**

Dear reviewers,

We thanks for your constructive feedback on our work. If you have had a chance to look at our responses, can you please let us know if it addresses your concerns? We are happy to provide any additional clarification as needed.

---

### Decision · Program_Chairs · 2024-07-10

**Decision:**

Accept

**Comment:**

The paper identifies the prevalence of attention sinks in Bert and Roberta, and hypothesises that they may lead to interference across tasks in continual learning set ups. To address this, the authors propose an approach to diversifying attention. Reviewers thought the work took an interesting new angle on continual learning, and demonstrated improvements in experiments The work could be further strengthened by using more recent mask language models, and evaluating on more tasks. Overall this is borderline, but leaning towards acceptance.